# Development of a complex multidisciplinary medication review and deprescribing intervention in primary care for older people living with frailty and polypharmacy

Eloise Radcliffe[1,2]*, Alejandra Recio Saucedo[3], Clare Howard[4], Claire Sheikh[5], Katherine Bradbury[2,6], Paul Rutter[7], Sue Latter[8], Mark Lown[1], Lawrence Brad[9], Simon D.S. Fraser[1,2], Kinda Ibrahim[1,2]

1 School of Primary Care, Population Sciences and Medical Education, Faculty of Medicine, University of Southampton, 2 NIHR Applied Research Collaboration ARC Wessex, University of Southampton, Southampton, United Kingdom, 3 School of Healthcare Enterprise and Innovation, Trials and Studies Coordinating Centre, National Institute of Health Research Evaluation, University of Southampton, Southampton, United Kingdom, 4 Health Innovation Wessex, Science Park, Chilworth, Southampton, United Kingdom, 5 Living Well Partnership, Southampton, United Kingdom, 6 School of Psychology, University of Southampton, Southampton, United Kingdom, 7 School of Pharmacy and Biomedical Sciences, Portsmouth University, Portsmouth, United Kingdom, 8 School of Health Sciences, University of Southampton, Southampton, United Kingdom, 9 Westbourne Medical Centre, Westbourne, Bournemouth, United Kingdom

* e.radcliffe@soton.ac.uk

## Abstract

### Introduction

Reducing polypharmacy and overprescribing in older people living with frailty is challenging. Evidence suggests that this could be facilitated by structured medication review (SMR) and deprescribing processes involving the multidisciplinary team (MDT). This study aimed to develop an MDT SMR and deprescribing intervention in primary care for older people living with frailty.

### Methods

Intervention development was informed by the Medical Research Council framework for complex intervention and behaviour change and implementation theories. Intervention planning included: 1) a realist review of 28 papers that identified 33 context-mechanism-outcome configurations for successful MDT SMR and deprescribing in primary care, 2) a qualitative study with 26 healthcare professionals (HCPs), 13 older people with polypharmacy and their informal carers. The intervention's guiding principles were developed and intervention functions proposed, discussed and refined through an iterative process in four online co-design stakeholder workshops.

### Results

The final version of the complex intervention consisted of five components: 1) Proactive identification of patients living with frailty and polypharmacy for targeted SMR using

**Data availability statement:** Bona fide researchers, subject to registration may request supporting data via University of Southampton repository https://doi.org/10.5258/SOTON/D3364 (or email research-data@soton.ac.uk)

**Funding:** This study is funded by the National Institute for Health and Care Research Applied Research Collaboration (ARC) Wessex (ARC004) awarded to KI. The views expressed in this publication are those of the authors and not necessarily those of the National Institute for Health and Care Research or the Department of Health and Social Care.

**Competing interests:** The authors have declared that no competing interests exist.

routinely collected primary care data; 2) HCPs' preparation using an evidence-based deprescribing tool to identify and prioritise high-risk medications for deprescribing; 3) Preparing patients and carers using a leaflet sent prior to SMR explaining the purpose of SMR and reasons for potentially stopping or changing medications; 4) Conducting a person-centred SMR face-to-face or by phone, tailored to patient/carer needs, involving other MDT members based on their expertise; 5) Tailored follow-up plans allowing continuity of care and highlighting signs and symptoms for patients and carers to monitor, and arranging follow-up through text, phone or face-to-face appointment.

## Conclusion

A complex MDT SMR and deprescribing intervention for older people living with frailty was developed to address multiple challenges to deprescribing. The use of rigorous methods and behaviour and implementation theories potentially maximises the intervention's feasibility, acceptability and successful implementation.

## Introduction

Nearly half of people in England aged 65 and over take five or more regular medicines [1], referred to as polypharmacy [2]. Polypharmacy can cause a significant but avoidable burden and source of harm for patients and places strain on healthcare systems [3]. Polypharmacy in older people is associated with increased potentially inappropriate medications (PIMs), referring to whether a drug is safe or unsafe in terms of its pharmaceutical properties and encompassing the assessment of older people's prescription medications in the context of multimorbidity, complex medication regimes, functional and cognitive status, treatment goals and life expectancy [4]. The risk of falls, cognitive impairment, functional decline, hospital admission and death are all increased by PIMs [5–7] and these impacts can be amplified in those living with frailty [8].

Polypharmacy can be managed through medicines optimisation, commonly defined as 'a person-centred approach to safe and effective medicines use, to ensure people obtain the best possible outcomes from their medicines' [9]. In the UK, recommendations suggest that people living with frailty and those with complex and problematic polypharmacy should receive a structured medication review [SMRs] annually by their primary care team [9,10]. The focus on frailty is due to the growing evidence that suggests frailty may have an impact on drug pharmacokinetics and pharmacodynamics, therapeutic efficacy and toxicity, although these factors may also play a role in the development of frailty [11]. An important aspect of medication review is deprescribing which involves tapering/dose reduction, stopping, or switching drugs with the goal of improving outcomes [12]. Deprescribing has been shown to be feasible and safe across a wide range of conditions, medications, settings and with the use of different deprescribing tools [13–18]. Deprescribing can lead to a reduction in polypharmacy and PIMs and for those living with frailty, can result in important benefits in relation to depression, function and frailty status [19,20].

Despite the international efforts to embed SMRs in routine practice, facilitated in the UK by the expansion in the number of clinical pharmacists working in primary care [21], there is considerable variation in the way SMRs are organised and implemented by general practices. Significant barriers to annual medication reviews and potential deprescribing for older patients living with frailty have been identified both from patient and informal carer perspectives, and from health care professional's [HCP] perspectives. Patients and informal carers, may be resistant to make any changes to their medicines due to the perceived benefits and fear

of the potential consequences of stopping them [22]. The time intensive nature of SMRs, the need for multiple appointments and shared decision-making [23], GPs' lack of time, increased workloads, concerns about stopping medicines within the context of a lack of policies and guidelines, the complexity of the healthcare system, and the complexity of patients who are most at risk, particularly those who are older, and living with multimorbidity and frailty, are some of the key barriers reported by primary care practitioners [24].

Involving other prescribers in reviewing medications has been suggested to address some of these barriers [25]. In the UK, clinical pharmacists and other HCPs including nurses and physiotherapists, are increasingly working as independent prescribers within a multidisciplinary primary care team to consult with and treat patients directly and more countries are moving towards this model [26]. Evidence indicates that multidisciplinary interventions and those involving pharmacists are effective in reducing inappropriate prescribing. However, this evidence mainly derives from a research context rather than a 'real' context in General Practice, and there is a lack of research on how this sharing of responsibilities could work effectively [3,27,28]. Our aim was to design a complex multidisciplinary medication review and deprescribing intervention in primary care, within the UK, for older people living with frailty and polypharmacy.

## Methods

This work is part of a larger programme of research on the development and iMplementation Of a multidisciplinary medication review and Deprescribing Intervention among Frail older people in primarY care (MODIFY). This study received ethical approval from the UK Health Research Authority (Research Ethics Committee reference 22/PR/0580).

The Medical Research Council's framework [29] for developing complex interventions underpinned the study. The framework recommends creating and iteratively refining a programme theory for an intervention that describes the key intervention components (i.e., content and delivery), its mechanisms of action (i.e., how an intervention is expected to lead to its effects), and its outcomes [30].

Theory-, evidence-, and a Person-Based Approach (PBA) [31] to intervention development was used to guide the development and refinement of the intervention. The first stage of intervention planning was to conduct a realist review and synthesis of the existing evidence on interventions for medication review and deprescribing in primary care involving multidisciplinary teams (MDTs). This was supported by a primary qualitative study and expert consultation with a multidisciplinary group of stakeholders and a Patient and Public Involvement (PPI) team, made up of three patient and carer representatives, who were involved throughout the study. This provided us with an in-depth understanding of the perspectives and potential barriers and facilitators of the interventions target group to maximise intervention uptake, adherence and outcomes. The findings from these exploratory studies informed the development of theory-based behavioural analysis and the intervention guiding principles, which in turn were used in designing the intervention that was tested and refined through key stakeholder workshops. The phases of the intervention development and refinement are described below in detail.

### Phase 1. Collating and analysing evidence

Phase 1 comprised: a) a realist evidence synthesis, and b) a qualitative study involving patients, carers and healthcare professionals (HCPs).

**a. Realist evidence synthesis.** We carried out a realist review and synthesis of the available published evidence to understand when, why, and how interventions for medication review

and deprescribing in primary care involving multidisciplinary teams (MDTs) work (or do not work) for older people. This was crucial to identify design features that are acceptable to target users, feasible to implement and are likely to be effective. The methods of this realist synthesis and detailed findings are described elsewhere [32].

**b. Qualitative study.** A qualitative study was conducted with 39 participants recruited from the Southeast of England to understand what makes multidisciplinary medication review and deprescribing work well in primary care for older people living with frailty and polypharmacy (see supporting informationfile 1 for topic guides). All participants were recruited between 1st September- 30th December 2022 and all provided written informed consent. The sample comprised 10 community-dwelling patients aged 65 and over living with frailty and taking five or more medications and three informal carers who participated in individual in-person interviews at their home (see Table 1). Participants were recruited through General Practices across the Southeast of England in both urban and rural areas with a range of socio-economic deprivation levels. In addition, a total of 26 HCPs working at these same General Practice participated in five focus groups (n=22) [(three in-person at the general practice and two online) and one-to-one online interviews (n=4). Each focus group was held with healthcare professionals (HCPs) from different disciplines working together at the same practice (see Table 2). The realist review and qualitative study were used to develop and refine a programme theory which formed the basis of the intervention guiding principles, as outlined in phase two.

## Phase 2. Intervention planning

Phase 2 comprised of a)] identification of potential behaviour change determinants using a model based in psychological theory and implementation theory based in sociological theory, and b) development of guiding principles.

## Behavioural analysis

In order to take a systematic and rigorous approach to ensure the intervention development was grounded in an in-depth understanding of the needs of stakeholders and potential

Table 1.  Patient and carer demographic characteristics (qualitative study).

| Participants | Mean age | Gender | Mean number of pre-scribed medications | Receiving help from a carer/relative with medicines |
|---|---|---|---|---|
| Patients (n=10) | 76 | Female= 4 Male=6 | 7.6 | Yes=4, No=6 |
| Carers (n=3) | 62 | Female=3 | – | – |

Table 2.  Health care professional characteristics (qualitative study).

| Professional Role | Number of participants |
|---|---|
| Clinical Pharmacist | 8 |
| GP | 7 |
| Advanced Nurse Practitioner | 4 |
| Frailty practitioner/coordinator | 3 |
| Medical students on placement | 2 |
| Dietician | 1 |
| Physiotherapist | 1 |
| **Total** | **26** |

challenges, we were guided by the Behaviour Change Wheel (BCW) [33,34]. At the centre of this framework is a 'behaviour system' involving three essential conditions for behaviour change: capability, opportunity, and motivation 'COM-B system'. Nine intervention functions are placed around the COM-B system aimed at addressing deficits in one or more of these three conditions for behaviour change; and around this are placed seven categories of policy or practice that could enable those interventions to occur. Based on the findings from phase 1, two team members (ER & KI) identified potential key behaviours (i.e., what needs to change for a behaviour to occur) to be targeted by the intervention. The identified key behaviours from phase 1 were first mapped onto the three COM-B subcomponents to provide a clear description of the intervention, then potential intervention functions were suggested based on the identified behaviours.

A further stage involved mapping the behavioural aspects onto the four constructs of the Normalisation Process Theory (NPT) [35] to ensure we had addressed any additional potential barriers to intervention success. NPT provides a conceptual framework for understanding and evaluating the processes by which complex interventions are routinely operationalised, embedded and sustained in everyday practice. NPT highlights four constructs 'coherence', 'cognitive participation', 'collective action' and 'reflexive monitoring', that are necessary for an intervention to be successfully implemented.

## Guiding principles

The next step was to develop a set of guiding principles [36] which were further refined through discussion with the wider research team, in order to maximise acceptability and feasibility. Drawing on the findings from phase 1 and the behavioural analysis in phase 2, the guiding principles consisted of 'the problem' i.e., key aspects of behaviour that the intervention needed to address, the pathway or stage of the medication review and deprescribing process, the intervention design objectives and the key 'intervention features' that will achieve these objectives.

## Phase 3. Intervention development and optimisation

Phase 3 involved the development of a preliminary intervention and support resources through four online stakeholder co-design workshops. The workshops were attended by three patients and carers (members of the PPI group), 10 HCPs working in primary care across the UK (clinical pharmacists, GPs, frailty nurses), recruited through the Clinical Research Network, and nine clinical and academic experts recruited through our own professional networks. During the workshops the intervention guiding principles, preliminary intervention content, format and delivery were presented and discussed. Data from workshops was analysed drawing on PBA. PBA aims to ground the development of behaviour change interventions in an understanding of the perspective and psychosocial context of the people who will use them, through iterative qualitative methods [37]. This helped to further refine the intervention using an iterative approach, to maximise acceptability and feasibility for patients and carers, and healthcare professionals delivering it.

## Results

## Phase 1. collating and analysing evidence

Based on a realist approach, evidence from 28 studies and interviews/focus groups with 39 primary care HCPs and patients and carers were analysed and a programme theory of 33 context-mechanism-outcome (CMOs) configurations was developed that describe key mechanisms for successful deprescribing in primary care. These CMOs were grouped under

four themes: 1) HCPs roles, responsibilities and relationships; 2) HCPs training and education; 3) the format and process of the medication review 4) involvement and education of patients and informal carers. Key mechanisms related to health care professionals and the MDT process included integration of pharmacists within the primary care team; supportive infrastructure with clearly defined MDT roles to facilitate collaboration and communication; use of MDT expertise as and when necessary, targeted HCP training on deprescribing and patient engagement, accessible and easy to use deprescribing tools, and systems to prioritise and target high-risk patients (e.g., frail, 10+ medications). Patient and carer engagement key mechanisms included patient's familiarity with the role of practice pharmacists and purpose of SMR; trusting relationships between patients and HCPs, face-to-face medication review facilitating better shared decision making, carer involvement where appropriate, a patient-centred, holistic approach, focus on patient goals and preferences, offering deprescribing as a "drug holiday" (that can be restarted at any time if necessary), or start with "quick wins" (referring to medications that are relatively simple to deprescribe but that the patient is likely to feel the benefit of), and planned follow-up.

## Phase 2. Intervention planning

The key aspects of behaviour change identified based on phase 1 and the guiding principles developed to address them are presented in Table 3. The table outlines 'the problem' i.e., key aspects of behaviour that the intervention needed to address, the pathway or stage of the medication review and deprescribing process, the intervention design objectives and the key 'intervention features' (or solution) that aimed to address the objectives and the corresponding BCW subcomponents and NPT constructs that the features map on to.

Identifying patients most at risk of medication harm for targeted medication review and deprescribing was a key objective of the intervention. This aimed to address the challenge (or 'problem') of a lack of standardised method or criteria across primary care for identifying older patients most at risk of harm from PIMs, in order to target limited resources to those most in need of a medication review and deprescribing. The key intervention feature (or 'solution') was to develop an appropriate and feasible search strategy for identifying and targeting older frail patients most in-need of an SMR using routinely collected data and primary care clinical records.

The second objective of the intervention was to facilitate patient and carer engagement and involvement in the SMR and deprescribing process, and encourage them to consider deprescribing when the harms of medications outweigh the benefits of medication, or when the benefits are not achieved. Challenges were identified concerning older patients with polypharmacy and their carers being unfamiliar with the role of clinical pharmacists and other healthcare staff in primary care, and a lack of information and understanding about the purpose of medication review and rationale for stopping or changing medicines. In addition, patients with polypharmacy and their carers may be reluctant to stop some medications if they are prescribed by secondary care specialist doctors or because they have concerns about the negative impact on their health if they stop taking it. To address these challenges, the key intervention feature (or 'solution') was to provide patients with key information before their SMR appointment. This information would focus on the role of the clinical pharmacist, the purpose of medication review, what to expect during the appointment, common reasons for changing or stopping medicines (making clear that is it not about costs), how to prepare for the appointment and take part in discussions and decisions by prompting them to consider their experience with medicines and any questions they may have about their medicines. The information resource would also encourage patients to discuss any concerns they have about their medicines with a carer or a family member and bring them to the medication review appointment, if they wish.

**Table 3. Guiding principles for the SMR and deprescribing intervention.**

| Problem/ key aspects of behaviour change to address (based on evidence from the realist review, qualitative study and expert consultation) | Pathway | Intervention design objectives | Solution/ key intervention features | Behaviour Change Wheel mapping (intervention functions) | NPT construct |
|---|---|---|---|---|---|
| • Variation across practices in criteria and methods used to identify and invite 'high risk' patients for SMR<br>• Unclear process for patients to book SMR (e.g., when stated on their prescription 'Your annual medicines review is due')<br>• Need to maximise use of limited resources to target SMR to patients who are most at risk/ in need/ (time and staff) | Identification of patients | • To identify and target SMR to patients who are most at risk | • Develop appropriate and feasible search strategy for practice administrative staff to apply to clinical record system. This will accurately and efficiently identify and target older people with frailty and polypharmacy, most in-need of an SMR. | Enablement, environmental restructuring | Collective action |
| • Older patients with polypharmacy and their carers may be unfamiliar with the role of clinical pharmacists and other primary care staff, and may be reluctant to discuss their medicines with any HCP other than a GP, and have a lack of trust in other HCPs.<br>• Patients lack information about the purpose of medication review and the rationale for stopping or changing medicines<br>• Patients with polypharmacy and their carers may be reluctant to stop some medications if prescribed by specialist doctors or due to concerns about the negative impact on their health | Preparation by patients/ carers before SMR | • To facilitate patient and carer motivation and involvement in SMR/ deprescribing process through increasing awareness of the clinical pharmacist role, purpose of medication review, rationale for stopping/ changing medicines<br>• To encourage patients to raise any questions about medicines. risks/ benefits and consider deprescribing<br>• To encourage patients to invite a family member/ carer to the SMR to provide support for deprescribing, if appropriate. | • Before SMR appointment, provide patients with written information (sent via post/text/ email) explaining that a clinical pharmacist or GP will conduct the SMR, the purpose of the SMR, what to expect during the appointment, common reasons for changing or stopping medicines, how to prepare for the appointment.<br>• The resource should reassure patients that no medicines will be altered without agreement between them and the HCP conducting the review<br>• The resource should encourage patients to discuss any concerns they have about their medicines with a carer or a family member and to bring them to the medication review appointment, if they wish. | Motivation, education | Coherence, Cognitive participation |
| • Lack of confidence to deprescribe among HCPs due to multiple factors including a lack of evidence on the safety of deprescribing, and perceptions that many patients' are unwilling to engage in deprescribing.<br>• HCPs experience challenges in choosing and accessing the appropriate deprescribing tools | Preparation by HCPs before SMR | • To increase HCPs confidence to approach deprescribing in the SMR process | • Provide HCP with training resources on evidence-based approaches to SMR that facilitate a patient-centred approach and shared-decision making (to increase patient involvement, aligning treatment goals with priorities).<br>• Provide easy-to-access, user-friendly deprescribing digital tool with available evidence in one place to help HCPs identify opportunities for deprescribing, when appropriate, including plans for tapering and monitoring. | Training, enablement | Coherence, cognitive participation |

*(Continued)*

**Table 3.** (Continued)

| Problem/ key aspects of behaviour change to address (based on evidence from the realist review, qualitative study and expert consultation) | Pathway | Intervention design objectives | Solution/ key intervention features | Behaviour Change Wheel mapping (intervention functions) | NPT construct |
|---|---|---|---|---|---|
| •Lack of GP time to conduct SMRs<br>•Building patient/ carer trust and ensuring continuity of care can be challenging for time-poor GPs working in a system with high levels of demand<br>•Patients may feel GPs do not have time to address concerns or questions about medicines<br>•Patient/ carer involvement/ engagement in SMR/ deprescribing process can be challenging for HCPs due to various reasons (including patients being reluctant to reduce or change a medicine due to concerns and fears about worsening symptoms, lack of awareness of the rationale for deprescribing and the possible risks of taking medicines as age and frailty increases). | **SMR/ deprescribing process** | •To facilitate a trusting relationship and shared-decision making between patient, carers and health care professionals conducting SMR | •SMR designed to be conducted by any prescribing HCP (e.g., Clinical pharmacist, advanced nurse practitioners, who is qualified to conduct a medication review as part of their role (flexible approach)<br>•The option of an SMR with an HCP who is not a GP should allow for longer appointments, giving patients more time and facilitating rapport and trust<br>•Mode of delivery of medication review to be tailored according to patients' needs (e.g., face-to-face, phone)<br>•SMR to begin with a focus on patient preferences and priorities.<br>•Deprescribing digital tool will aid HCPs in their discussions with patients about which medicines should be deprescribed and the possible risks and benefits of continuing to take these medicines vs reducing or stopping them<br>•Provide training material to encourage HCP to consider the following approaches to deprescribing to engage patients and facilitate trust:<br>- offering deprescribing as a trial off medication, ('drug holiday') that can be monitored and restarted anytime if needed to increase patient engagement and trust<br>- Start with changing one medication at a time<br>- Start with simple deprescribing changes, tailored to the individual patient, that could lead to noticeable improvements in symptoms by patients ('quick wins'). | Enablement | Collective action |

*(Continued)*

**Table 3.** (Continued)

| Problem/ key aspects of behaviour change to address (based on evidence from the realist review, qualitative study and expert consultation) | Pathway | Intervention design objectives | Solution/ key intervention features | Behaviour Change Wheel mapping (intervention functions) | NPT construct |
|---|---|---|---|---|---|
| • Lack of MDT working for peer support and shared responsibility that could help to increase confidence in making deprescribing decisions • Lack of clear MDT process • Need to maximise efficient use of MDT resources and expertise for conducting SMR | MDT process and communication | • To support practices to establish clearly defined roles of the MDT in the SMR/ deprescribing process • Facilitate opportunities to consult with other team members as and when necessary | • Ensure there is a clearly nominated HCP who will conduct the SMRs • HCPs conducting the SMR are encouraged to consult with other members of the MDT as and when needed (e.g., patients with complex issues) using flexible communication methods (e.g., In-person, digital) • Provide easy-to-access, user-friendly deprescribing tool to identify opportunities for deprescribing, when appropriate, with available evidence in one place. This can facilitate discussions with other MDT members when necessary. | Enablement | Collective action |
| • Patient concern about lack of follow-up to monitor any potential worsening of symptoms or other adverse effects after changing, reducing or stopping medicines | Follow-up | • To facilitate patient engagement trust, and continuity of care in relation to the SMR and deprescribing process through an agreed follow up and monitoring plan | • Robust follow-up plans should be agreed with patients, allowing continuity of care and support for patients who are concerned about any changes made to their medicines. • Plans to include monitoring of symptoms/ side-effects, tailored to patients' needs and may involve follow-up telephone or in-person appointments with a focus on the provision of support in relation to their priorities. • Provide a template to document medication management plan for patient/ carers and scheduled follow-up (e.g., phone/ face-to-face), if appropriate, and a clear HCP point of contact for the patient, if needed. | Enablement | Reflexive monitoring |

The third objective was to increase HCPs confidence in deprescribing decision making. Factors identified as barriers to HCP confidence to deprescribe included a lack of evidence on the safety of deprescribing, perceptions that many patients are unwilling to engage in deprescribing, and difficulty choosing and accessing appropriate deprescribing tools. To address this, two key intervention features were proposed: HCP training resources on evidence-based approaches to SMR that facilitate a patient-centred approach and shared-decision making; and provide an easy-to-access, user-friendly deprescribing tool(s) with available evidence in a single place to help HCPs identify opportunities for deprescribing, when appropriate, including plans for tapering and monitoring.

The fourth objective was to facilitate a trusting relationship and shared-decision making between patients, carers and health care professionals. Challenges were identified for HCPs regarding patient and carer involvement and engagement in the SMR/ deprescribing process and building patient and carer trust. Key intervention features were a flexible approach, with SMRs tailored according to patients' needs (e.g., face-to-face, phone), to be conducted by any prescribing HCP, beginning with a focus on patient preferences and priorities and HCP training materials that encourage approaches to deprescribing that engage patients and facilitate trust.

The fifth objective was to establish clearly defined roles of the MDT in the SMR/ deprescribing process to facilitate opportunities to consult with other team members as and when necessary. Challenges were identified concerning the need for a clear process of communication between MDT members and the ability to access support from other MDT members for complex deprescribing decisions, to facilitate HCP confidence. To address this, the key intervention features were to ensure there is a clearly nominated HCP who will be responsible for conducting the SMRs, and encourage them to consult with other MDT members as and when necessary, facilitated by an easy-to-access, user-friendly online digital tool to identify opportunities for deprescribing when appropriate.

The final objective was to facilitate patient engagement, trust, and continuity of care in relation to the SMR and deprescribing process through an agreed follow up and monitoring plan. The evidence identified patient concern about lack of follow-up and continuity of care. In order to address these challenges the intervention encourages robust follow-up plans to be agreed with patients and tailored to their needs, allowing continuity of care and support for patients who are concerned about any changes made to their medicines, and may involve follow-up telephone or in-person appointments. Another intervention feature was a template to document a medication management plan for patient/ carers and scheduled follow-up (e.g., phone/ face-to-face), if appropriate, and a clear HCP point of contact for the patient, if needed.

## Phase 3. Intervention development and optimisation

Based on the guiding principles identified in phase 2 we developed a preliminary intervention for SMR and deprescribing in primary care with a multidisciplinary approach, together with support resources. The intervention guiding principles, preliminary intervention content, format and delivery were presented and discussed during four online stakeholder workshops and data from workshops was analysed drawing on PBA. This helped to further refine the intervention using an iterative approach, to maximise acceptability and feasibility for patients and carers, and health care professionals delivering it. Table 4 gives an overview of the intervention content discussed and any changes agreed on, mapped onto the COM-B subcomponents and NPT constructs. The final intervention is presented in Fig 1, and consists of five main components/stages:

**Table 4. Overview of intervention content discussed and agreed on during stakeholder workshops.**

| Intervention component | Intervention content discussed at stakeholder workshops | Final version of intervention agreed | Behaviour change wheel mapping (intervention functions) | NPT construct |
|---|---|---|---|---|
| Identification of high-risk patients | Criteria for identification of high-risk patients<br>•Age: 65 and over vs 75 and over?<br>•Frailty: which measure is it best to use- Rockwood Clinical Frailty Scale vs electronic Frailty Index (eFI)?<br>•Number of regular medicines: 5 or more vs 10 or more?<br>•Best methods to identify patients? E.g., practice database search or search using NHS Business Services Authority national polypharmacy comparator datasets developed by the National Health Innovation Network, mapped onto frailty scores? | To identify those likely to be at greater risk, the intervention will be targeted to patients who are:<br>•Aged 75 and over<br>•Taking 10 or more regular medicines<br>•Identified as having moderate to severe frailty based on eFI as this data is available on all general practices databases, unlike other measures.<br>•Practice database searches (SystemOne and EMIS) as easier and quicker to implement by practice staff. | Enablement, environmental restructuring | Collective action |
| Patient and carers preparation for medication review | How best can we facilitate patient involvement and engagement in the SMR and deprescribing process?<br>Should we use an existing resource to inform patients and carers about the purpose of SMR and reasons for potentially stopping or changing medications and encourage them to prepare and ask questions about medicines? If so, which one? Or develop our own resource?<br>How is it best to encourage carer involvement in the SMR, where appropriate? | We will use an existing resource developed and evaluated by the University of Leeds and Bradford Teaching Hospital NHS Foundation Trust [38–40]. This resource is recommended for use by other organizations (e.g., Age UK and the Health Innovation Network) ('Reviewing your medicines' Resources to support patients having a Structured Medication Review - The Heath Innovation Network [thehealthinnovationnetwork.co.uk]).<br>This leaflet encourages patients to discuss their medicines with their carers and invite them to the SMR, where appropriate.<br>Each patient will be sent the leaflet [by post, text or email] before their SMR appointment. | Motivation, education | Coherence, cognitive participation |
| Primary care staff preparation for medication review | We proposed to use the Patient Attitudes Towards Deprescribing (PATD) questionnaire [41] to identify patients who may be more open and amenable to deprescribing, before the SMR. | Feedback on use of the PATD questionnaire indicated this would be difficult to implement before the SMR and would be cumbersome and time consuming for patients and HCPs. Therefore the decision was taken not to use the PADT as part of the intervention. | Enablement | Coherence |
| Primary care staff preparation for medication review | What is the best way to provide support, training and education to primary care staff conducting medication reviews and deprescribing?<br>How best can we encourage MDT working on medication reviews and deprescribing?<br>How best can we facilitate HCPs in involving and engaging patients in the SMR and deprescribing process?<br>We proposed the use of a recently developed online tool developed by PrescQIPP (a community interest company, helping NHS organisations to improve medicines-related care to patients). The Improving Medicines and Polypharmacy Appropriateness Clinical Tool (IMPACT) which identifies clinical and deprescribing priority with recommendations and considerations for appropriately continuing or stopping medicines. [https://www.prescqipp.info/our-resources/bulletins/bulletin-268-impact/]. It comprises of evidence and recommendations collated from a wide range of resources and tools. All general practices and primary care staff have access to the tool.<br>Would HCPs find this tool useful?<br>Would using this tool be feasible in general practice?<br>What training would be needed to use the tool? | We will provide deprescribing tip sheets based on our published realist review of the available evidence [32]. The 3–4 evidence-based tip sheets on deprescribing will focus on summarising the evidence on deprescribing in this population, how to work as an MDT to support decisions about medicines, top tips on discussing stopping medicines with patients and carers and the need for follow up plans.<br>After demonstrating The PrescQIPP Improving Medicines and Polypharmacy Appropriateness Clinical Tool (IMPACT) and gaining positive feedback from the majority of HCPs during the stakeholder workshops we decided to use the tool as a key aspect of our intervention.<br>Feedback in the workshops included the potential benefits of using the IMPACT tool to facilitate HCP confidence in making decisions with patients about medicines to be deprescribed and facilitate MDT working when appropriate, by producing an individualised patient-specific PDF of medications to identify and prioritise high-risk medications for deprescribing. A number of HCPs commented that a patient-friendly version of the PDF would be helpful, and it would save time if the IMPACT tool was embedded within the practice IT system, rather than having to log on to a separate tool. Overall feedback on the IMPACT tool was very positive and it was felt it would be particularly useful for more junior clinical pharmacists.<br>A brief training video (already developed by PrescQIPP) will be shown to primary care (and made available after) about how to access and use the tool.<br>A sheet demonstrating how to register with PrescQIPP will also be provided. | Training, education Enablement, motivation | Coherence Coherence, cognitive participation |

*(Continued)*

**Table 4.** (Continued)

| Intervention component | Intervention content discussed at stakeholder workshops | Final version of intervention agreed | Behaviour change wheel mapping (intervention functions) | NPT construct |
|---|---|---|---|---|
| Medication review appointment | Who should conduct the SMR?<br>How should the SMR be conducted? Eg. In-person, over the phone? | The SMR will be conducted by an appropriate HCP, as agreed by the practice. This could include a clinical pharmacist, GP or advanced nurse practitioner, frailty team staff.<br>The SMR will ideally be conducted face-to-face, however phone or video consultation could be used, based on patient's preferences and needs (flexible). | Enablement | Cognitive participation, collective action |
| Patient follow-up | We proposed to use a template to allow HCPs to give the patient a written record of any changes that may have been made to the patients' medication. This should include any changes in medication doses, tapering protocols, symptoms to monitor, instructions for restarting medications if needed. Plans for follow-up and monitoring of symptoms should also be agreed with the patient/carer and recorded on the template provided. Should we use an existing resource or develop our own template?<br>What is the best way for HCPs to provide follow-up support to patients? (e.g., phone call, text message, in-person scheduled appointment or request that the patient contacts the practice if necessary). | Use existing resource developed and evaluated by the University of Leeds and Bradford Teaching Hospital NHS Foundation Trust [38–40], as above for the patient preparation resource. This leaflet is recommended for use by other organizations [e.g., Age UK and the Health Innovation Network].<br>('Safely stopping your medicines' Resources to support patients having a Structured Medication Review The Heath Innovation Network [thehealthinnovationnetwork.co.uk])<br>Follow-up support will be tailored to the individual patient and carer and their medicines [flexible]. | Enablement | Reflexive monitoring |

## 1. Identification of high-risk patients.

A search of practice clinical records to be used to identify patients aged 75 and over, taking 10 or more medicines, with moderate to severe frailty based on electronic Frailty-Index (eFI score =>0.25) [42] to identify those with moderate-severe frailty. The search excluded those in care homes, those at the end of life, those without the capacity to give informed consent and those receiving an SMR within the past 6 months.

## 2. Patients and carers preparation.

Patients and carers who have agreed to attend an SMR will be sent (by post, text or email) written information, using an existing available template developed by other researchers based in the UK and approved by the NHS, (Reviewing your medicine' Resources to support patients having a Structured Medication Review - The Heath Innovation Network thehealthinnovationnetwork.co.uk)] about the purpose of SMR and reasons for potentially stopping or changing medications and encourage them to prepare and ask questions about medicines. Patients are encouraged to discuss with family members or carers and to bring them to the appointment, if they wish.

**3. HCPs preparation for conducting an SMR.** Deprescribing Tip sheets (see supporting information file 2): Five evidence-based deprescribing tip sheets were developed by the research team to summarise the evidence on deprescribing in this population in an easy to read format, how to work as an MDT to support decisions about medicines, top tips on discussing stopping medicines with patients and carers such as considering "quick wins" and "drug holidays", and the need for follow up plans.

**Fig 1. Summary of final intervention content.**

The PrescQIPP Improving Medicines and Polypharmacy Appropriateness Clinical Tool (IMPACT) [43] was chosen and agreed by stakeholders in the workshops to be used as part of the intervention. It is a digital tool that comprises of evidence and recommendations collated from a wide range of deprescribing resources and tools including STOPP-START [44], STOP-Frail [45], NICE guidelines [9] and the Canadian medication appropriateness and deprescribing network website [46]. The tool prioritises medicines that could be deprescribed, with recommendations and considerations for appropriately continuing or stopping medicines and produces an individualised patient-specific list of high-risk medications for deprescribing to inform decisions and discussions with patients. A brief training video and a sheet demonstrating how to register with PrescQIPP were to be provided to HCPs in primary care.

Understanding patient context and individual circumstances: HCPs responsible for SMRs should review the patient' medical records for any relevant information such as tests/scans, hospital admissions, reasons and indications of drugs. They also should consult other HCPs within primary care (e.g. GPs) or secondary care (specialist practitioner) as appropriate and when needed to gather further details or to discuss any complex issues.

**4. Conducting a person-centred SMR.** An appropriate prescribing HCP will conduct the SMR, including a clinical pharmacist, GP or advanced nurse practitioner, as agreed by the practice. This could be conducted face-to-face, however phone or video consultation could be used, based on patient's circumstances. During the appointment, HCPs conducting SMR should:

- Focus on patient's and carer's needs, preferences and goals (person-centred approach). Ask "what matters most to you?" and identify any issues related to medication management. For example, use the "Reviewing your medicines" as an opportunity to empower patients to ask questions about their medicines.

- Identify opportunities for deprescribing (stop, reduce dose, switch), and use IMPACT tool to highlight the high-risk medications they are prescribed

- Use the tips and questions provided in the deprescribing tip sheets to facilitate discussions.

- Consider "Quick wins" and "drug holidays" when offering to stop a medicine.

- Involve family members/ carers in decision making and educate them of the risks and benefits of medicines.

- Keep a record of these discussions on patients records so other HCPs are aware of any changes to medicines.

**5. Tailored follow-up.** Using an existing template developed by researchers and commissioned by NHS England (Resources to support patients having a Structured Medication Review The Heath Innovation Network (thehealthinnovationnetwork.co.uk)], the HCP will provide the patient with a written record of any changes that may have been made to medications including any signs or symptoms to look for and monitor when coming off a medication and agree plans for follow-up appointments or referrals.

## Discussion

This original research article describes the development of a complex multidisciplinary medication review and deprescribing intervention for primary care, targeting older people living with frailty and polypharmacy, using a theory-, evidence-, and person-based approach drawing on behaviour change and implementation theories. These different approaches provided complementary insights to maximize the intervention's acceptability, feasibility of delivery,

and likely effectiveness. This article provides a comprehensive description of the intervention planning process to allow other researchers to easily understand how this methodology could be applied to different intervention contexts. The comprehensive intervention description will also facilitate intervention replication and evaluation, and comparison with different medication review and deprescribing interventions.

The focus on people living with frailty and polypharmacy is a significant strength of our intervention. In this patient population, the process is complex and challenging due to dynamic changes in frailty, difficulties assessing risks and benefits from single long term condition guideline prescribing recommendations, increased sensitivity to adverse drug reactions, and misalignment of therapeutic goals with patient's priorities [47]. It has been argued that HCPs need to appreciate that evidence-based practice in deprescribing is an 'ongoing interpretive process that includes the patient's story, scientific story and professional story' [47]. Our intervention employs a person-centred approach, looking through the lens of frailty and/or multimorbidity and considers how the patient's medicines affect their function and quality of life as their circumstances change. Based on a review of the evidence, we have previously argued that describing should be approached as a continuum, beginning when a prescription is first initiated, rather than a one-off process [32]. Regular SMR and clear follow-up plans after deprescribing are key, and are aspects that we have aimed to address in our intervention.

There is some evidence from a systematic review that deprescribing among those living with frailty is feasible, safe, and can lead to improvement in cognition, depression, and frailty status [19]. Only two deprescribing studies were implemented in primary care focusing on people with frailty. One was a quasi-experimental pretest-post-test design in Canada among 54 older frail patients with polypharmacy which used a pharmacist-led medication review using STOPP-START [48]. There were no significant changes in total number of medications taken by patients before and after, but the intervention significantly decreased numbers of inappropriate medications [1.15 meds pre to 0.9 meds post; $P$ =.006]. The second study was a cluster RCT in Germany among 521 patients where GPs in the intervention group received a complex intervention involving three training sessions on family conferences, a deprescribing guideline, and a toolkit with relevant nonpharmacologic interventions [49]. The intervention did not achieve sustainable effects in reducing the number of hospitalizations or the number of medications and PIMs after 12 months. An important limitation of both studies was not including stakeholders, patients and the public in designing the intervention, and neither evaluated patients' experiences of the intervention. Using qualitative research alongside PPI in our research allowed us to capture a more diverse set of views and experiences that needed to be addressed to make the intervention more clinically relevant, accessible and implementable.

The intervention is currently being tested in a feasibility study across five general practices in Southeast England, together with a qualitative process and health economic evaluation. The outcome measures were discussed and agreed with the PPI group and key academic and clinical stakeholders. These include medication-related outcomes (number of medications pre and post intervention, types of deprescribing recommendations), clinical-related outcomes (ncluding treatment burden, frailty status, falls), quality of life, healthcare use and cost related outcomes, and safety outcomes (hospitalization and mortality rates).

Deprescribing is complex and should consider the individual context and embrace patient centredness and collaborative teamwork, including patients, relatives and/or caregivers [47]. Multiple deprescribing interventions have been evaluated, but there has been limited consideration of implementation factors in these previous deprescribing interventions, especially with regard to the personnel and resources in existing health-care systems and the feasibility of incorporating components of deprescribing interventions into the routine care processes of clinicians [50]. Our complex intervention could potentially address these gaps as it was

designed based on detailed behavioural analysis informed by behaviour change and implementation theories to take account of the wider social and system-based contexts, alongside extensive stakeholder engagement including those who will both receive and deliver the intervention. Our rigorous methods for intervention design have aimed to maximise the feasibility of the intervention, to ensure it is an SMR process that can be implemented within existing primary care systems, with the potential for long-term sustainability. It brings together resources, including the IMPACT tool and patient information resources, that are freely available to all NHS health care professionals, and are already in use within the NHS. We have also ensured that any prescribing HCP can implement the SMR and deprescribing process, so clinical pharmacists can lead the process, with an MDT approach enabling other HCPs to be involved as and when needed. This is in clear alignment with UK health policy, advocating the role of clinical pharmacists within a multidisciplinary approach to SMR in primary care to address inappropriate polypharmacy [51] and in alignment with wider UK policy aiming to address overprescribing [52]. Our intervention aims to make the SMR and deprescribing process more efficient and standardized within routine care, with the potential to save time and resources, in addition to improving patient outcomes. In addition to a feasibility study, we are currently conducting policy engagement work to consult with policymakers to identify facilitators to implementation and integration of the intervention within the workflow of primary care teams, but also beyond to community settings.

The intervention has been designed to utilise the expertise of other prescribing practitioners in medication-related issues and to address the time constraints and workload barriers of general practitioners, identified in our qualitative study and reflected in the findings from other recent qualitative research [23] Our realist review has identified the key role of co-located, well integrated GP-based clinical pharmacists in supporting medication reviews, in line with the literature [53]. Co-located integration of pharmacists has been shown to deliver a range of non-dispensing interventions, with medication management reviews being a primary activity. Pharmacist integration may also reduce GP workload directly through support for medication-related administration and management, medications reconciliation following hospital discharge and indirectly though reducing medication related adverse events leading to emergency department attendance and hospitalizations [54].

Despite HCPs' awareness of the value and need for deprescribing medications, lack of confidence in making these decisions and the difficulty in choosing appropriate deprescribing tools to support these decisions have hindered implementing deprescribing in clinical practice. There are hundreds of drug-specific or generic deprescribing tools, frameworks and algorithms available, making it very challenging to frontline clinicians to identify which one to use and how to access them during their busy clinical practice [16,55]. Our intervention has aimed to address this through the use of an online digital tool to prioritise medicines for deprescribing and provide evidence in one easily accessible place and through a multidisciplinary aspect to allow collaboration in decision-making with other health professionals. Using the online PrescQIPP IMPACT tool within the context of a person-centred SMR to aid discussions with patients and carers, can potentially facilitate involvement in shared decision-making about their medicines and improve HCPs' confidence in deprescribing decisions.

There is the potential for the use of Artificial Intelligence (AI) in decision support tools for SMRs. Tools that can guide prescribing and deprescribing decisions already exist and have the potential to bring real benefits, but caution has been urged regarding the potential to cause very significant harms to patients, with the 'true impact' of these tools on prescribing decisions being very hard to estimate [56]. This is a growing area of research and it is argued that in future significant efforts should be made, in collaboration with data scientists,

to better assess the value of AI-powered apps and tools for clinical pharmacy in real practice [57]. While AI is likely to play a significant role in future, in guiding decisions around deprescribing, the shared decision-making process between HCPs and patients and carers should always be central in the deprescribing process, with a trusting relationship between HCPs and patients identified as playing a key role [32,58].Our study identified the role that patient attitudes play in deprescribing, often reflecting wider cultural attitudes towards medicines, and the need to inform and educate patients to facilitate engagement in discussions about their medicines. Our qualitative study identified that patients were often unsure about the purpose of an SMR and unfamiliar with the role of the clinical pharmacist, reflecting other qualitative findings [59]. We aimed to address this in our intervention with the use of previously developed and evaluated resources to inform patients about SMRs, reasons for possible deprescribing and to encourage them to ask questions about their medicines [38–40]. We identified the key role of trusting relationships between patients and HCPs during the medication review process and in being open to deprescribing recommendations, also emphasised by the TAILOR evidence synthesis [58]. Research has shown variation in the degree of trust that patients of different races and ethnic backgrounds have in the healthcare system. One study among people living with heart failure and frailty showed that non-white participants were less likely to express agreement with wanting to reduce their number of medicines and authors attributed this to a lack of trusting relationship with their clinicians [60]. We have aimed to address this through a number of intervention functions including the emphasis on preparation for both HCPs delivering SMR and patients and their carers prior to the appointment, employing a person-centred approach that focuses on patients' preferences and needs, and providing tailored follow-up support and contact for patients to facilitate trust and continuity of care.

## Strengths and limitations

A systematic approach to intervention design using an integrated theory-, evidence-, and a person-based approach has been used. There are alternative approaches to PBA and the BCW that could have been used in developing this intervention, for example intervention mapping [61], however this has been critiqued for being time-consuming and overly prescriptive [62,63]. NPT is unique in offering a clear theoretical framework that addresses both individual and organisational level factors needed for successful implementation and integration of interventions into routine work (normalisation) [64]. The strength of NPT compared to other approaches lies in the recognition of the wider system and that healthcare is a collective activity requiring a multitude of interactions between professionals, patients, managers and others [64]. In employing a behaviour change model combined with NPT implementation theory, to take account of the wider system and social context of the intervention, we ensured a consistent and rigorous approach to intervention development to maximise the intervention's acceptability, feasibility and potential effectiveness.

Patients, carers and HCPs in our qualitative study and stakeholder workshops were recruited from UK general practices in areas that were both urban and rural, with a range of socio-economic deprivation levels, with similar numbers of male and female, and with primary care HCPs from various disciplines and with a range of years of professional experience. However, there was a lack of ethnic and cultural diversity in our sample, with the majority from White British backgrounds which may limit the generalisability of the findings to non-white populations. However, our PPI group, involved in all stages of the intervention development process, and subsequently in the current feasibility study, is a diverse group in terms of gender, age and ethnicity. The study was conducted in the UK where the primary health care setting, context, and workforce might differ from other countries limiting the transferability

of the intervention. Another limitation is that the intervention had to be designed within the constraints of the wider health care system and available resources that GP practices have access to, to ensure it was feasible, acceptable and within the scope of our study. Therefore, although we were able to address many barriers identified throughout the design process, we were not able to address all barriers. For example wider issues in communication and information sharing between primary and secondary care were challenges, also identified in other recent research [23], that could not be addressed within the timeline and scope of the study. We recommend that future research focuses on understanding and addressing these complex system-level challenges.

## Conclusions and implications for research and practice

A complex MDT medication review and deprescribing intervention for older people living with frailty was developed to address multiple barriers for both patients and HCPs. The use of rigorous methods and drawing on behaviour and implementation theories helped to identify key mechanisms for successful deprescribing in clinical practice, many of which were included in the co-developed intervention, potentially maximising the intervention's feasibility, acceptability and implementation. The way in which SMRs are currently implemented in primary care in the UK is very variable and is likely impacted by the increasing demand on primary care services. Our intervention has the potential to improve and standardise the SMR and deprescribing process and consequently improve outcomes for vulnerable older patients living with frailty and polypharmacy. The intervention is currently being tested in a feasibility study, with the aim of conducting a full randomised controlled trial in future before moving to potential adoption in clinical practice.

## Supporting information

**S1. MODIFY qualitative topic guides.**
(PDF)

**S2. Health care professional deprescribing tip sheet.**
(PDF)

## Acknowledgments

We would like to thank our PPI group and all patients, carers and health care professionals who participated in the study.

## Author contributions

**Conceptualization:** Clare Howard, Claire Sheikh, Katherine Bradbury, Paul Rutter, Sue Latter, Mark Lown, Lawrence Brad, Simon D.S. Fraser, Kinda Ibrahim.

**Data curation:** Eloise Radcliffe, Alejandra Recio Saucedo, Clare Howard, Claire Sheikh, Katherine Bradbury, Paul Rutter, Sue Latter, Mark Lown, Lawrence Brad, Simon D.S. Fraser, Kinda Ibrahim.

**Formal analysis:** Eloise Radcliffe, Alejandra Recio Saucedo, Clare Howard, Claire Sheikh, Katherine Bradbury, Paul Rutter, Sue Latter, Mark Lown, Lawrence Brad, Simon D.S. Fraser, Kinda Ibrahim.

**Funding acquisition:** Alejandra Recio Saucedo, Clare Howard, Claire Sheikh, Katherine Bradbury, Paul Rutter, Sue Latter, Mark Lown, Lawrence Brad, Simon D.S. Fraser, Kinda Ibrahim.

**Investigation:** Eloise Radcliffe, Clare Howard, Claire Sheikh, Katherine Bradbury, Paul Rutter, Sue Latter, Mark Lown, Lawrence Brad, Simon D.S. Fraser, Kinda Ibrahim.

**Methodology:** Eloise Radcliffe, Alejandra Recio Saucedo, Clare Howard, Claire Sheikh, Katherine Bradbury, Paul Rutter, Sue Latter, Mark Lown, Lawrence Brad, Simon D.S. Fraser, Kinda Ibrahim.

**Project administration:** Eloise Radcliffe, Kinda Ibrahim.

**Resources:** Kinda Ibrahim.

**Supervision:** Kinda Ibrahim.

**Writing – original draft:** Eloise Radcliffe, Kinda Ibrahim.

**Writing – review & editing:** Eloise Radcliffe, Alejandra Recio Saucedo, Clare Howard, Claire Sheikh, Katherine Bradbury, Paul Rutter, Sue Latter, Mark Lown, Lawrence Brad, Simon D.S. Fraser, Kinda Ibrahim.

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
