## [Decision Letter · Decision Letter 0]

15 Dec 2024

PONE-D-24-53398Development of a complex multidisciplinary medication review and deprescribing intervention in primary care for older people living with frailty and polypharmacyPLOS ONE

Dear Dr. Radcliffe,

Thank you for submitting your manuscript to PLOS ONE. After careful consideration, we feel that it has merit but does not fully meet PLOS ONE’s publication criteria as it currently stands. Therefore, we invite you to submit a revised version of the manuscript that addresses the points raised during the review process.

in particular, in responding to the reviewers comments, consider any limitations of the study, for example, the lack of discussion on the scalability of the intervention from the pilot setting.

We look forward to receiving your revised manuscript.

Kind regards,

Kathleen Bennett

Academic Editor

PLOS ONE

2. Please describe in your methods section how capacity to provide consent was determined for the participants in this study. Please also state whether your ethics committee or IRB approved this consent procedure. If you did not assess capacity to consent please briefly outline why this was not necessary in this case.

“This study is funded by the National Institute for Health and Care Research Applied Research Collaboration (ARC) Wessex.”

4. In this instance it seems there may be acceptable restrictions in place that prevent the public sharing of your minimal data. However, in line with our goal of ensuring long-term data availability to all interested researchers, PLOS’ Data Policy states that authors cannot be the sole named individuals responsible for ensuring data access (http://journals.plos.org/plosone/s/data-availability#loc-acceptable-data-sharing-methods).

Reviewers' comments:

Reviewer's Responses to Questions

**Comments to the Author**

1. Is the manuscript technically sound, and do the data support the conclusions?

Reviewer #1: Yes

Reviewer #2: Yes

2. Has the statistical analysis been performed appropriately and rigorously? 

Reviewer #1: Yes

Reviewer #2: N/A

3. Have the authors made all data underlying the findings in their manuscript fully available?

Reviewer #1: Yes

Reviewer #2: Yes

4. Is the manuscript presented in an intelligible fashion and written in standard English?

Reviewer #1: Yes

Reviewer #2: Yes

5. Review Comments to the Author

Reviewer #1: Intervention design and real-world application: While the manuscript outlines a comprehensive approach to developing a multidisciplinary medication review intervention, it lacks sufficient discussion on potential barriers to scaling the intervention beyond the pilot settings. A detailed plan for addressing resource constraints in diverse healthcare systems could enhance its utility.

Stakeholder diversity in development process: The study involved stakeholders, including patients, caregivers, and healthcare professionals, but the representation of certain groups, such as minority ethnic populations or those from varied socioeconomic backgrounds, is unclear. Expanding on the demographic diversity of participants could strengthen the intervention's generalizability.

Outcome measures and feasibility of follow-up plans: Although the manuscript emphasizes tailored follow-up plans for continuity of care, it does not specify measurable outcomes for evaluating the effectiveness of these follow-ups. Including specific metrics, such as adherence rates or health outcomes post-intervention, would provide a more robust evaluation framework.

Ethical considerations in data sharing: The manuscript mentions that anonymized qualitative data is available upon request but does not elaborate on the ethical safeguards for participant confidentiality. Further clarity on how sensitive patient data will be protected during broader implementation is necessary.

Behavior change model justification: The manuscript employs the Behaviour Change Wheel (BCW) and Normalization Process Theory (NPT) frameworks but does not sufficiently justify why these models were chosen over others. A comparative discussion could enhance understanding of their appropriateness for this intervention.

Evaluation of deprescribing tools: While the use of tools like the IMPACT tool is highlighted, there is limited discussion on their limitations or feedback from healthcare professionals during the stakeholder workshops. Addressing these could inform future iterations of the intervention.

Sustainability and long-term implementation: The manuscript outlines a structured intervention development process but lacks a clear strategy for long-term sustainability. Details on funding models, policy alignment, or integration into existing healthcare workflows would strengthen the argument for its widespread adoption.

Reviewer #2: This is a very interesting paper with some highly useful recommendations for improving the process of SMRs particularly in those with multimorbidity and polypharmacy. It is well written, the qualitative evidence gathereed to support the findings is robust and recommendaitons are sound. I enjoyed reading the paper.

The issues around barriers and facilitators in SMRs have recenrtly been reviewed extensively in another PLOS One paper - see Abuzour AS, Wilson SA, Woodall AA, Mair FS, Clegg A, Shantsila E, Gabbay M, Abaho M, Aslam A, Bollegala D, Cant H, Griffiths A, Hama L, Leeming G, Lo E, Maskell S, O'Connell M, Popoola O, Relton S, Ruddle RA, Schofield P, Sperrin M, Staa TV, Buchan I, Walker LE. A qualitative exploration of barriers to efficient and effective structured medication reviews in primary care: Findings from the DynAIRx study. PLoS One. 2024 Aug 30;19(8):e0299770. doi: 10.1371/journal.pone.0299770. PMID: 39213435; PMCID: PMC11364411, some comparison and reflection between the findings of this team and those of the Southampton group would be helpful. Also, opportunities to use decision support tools (eg AI) in supporitng SMRs are highly relevant and while not explored directly in this paper, some discusison of such methodology in the discussion section should be made I feel to refflect this growing area of research interest and practice in the diccussion section. Overall however I few this as a highly valuable addition to the research evidence in this field.

6. PLOS authors have the option to publish the peer review history of their article (what does this mean? ). If published, this will include your full peer review and any attached files.

**Do you want your identity to be public for this peer review?** For information about this choice, including consent withdrawal, please see our Privacy Policy .

Reviewer #1: **Yes: ** Saibal Das

Reviewer #2: No

---

## [Author Response · Author response to Decision Letter 1]

29 Jan 2025

Editorial Office

PLOS ONE

16th January 2025

Dear Reviewers,

Many thanks for taking the time to review our manuscript, entitled “Development of a complex multidisciplinary medication review and deprescribing intervention in primary care for older people living with frailty and polypharmacy”. Thank you for all of the very constructive feedback, which we have now addressed, further strengthening the paper. Below we have outlined how we have addressed each of the comments (also in the attached letter in table format).

Reviewer #1:

Intervention design and real-world application: While the manuscript outlines a comprehensive approach to developing a multidisciplinary medication review intervention, it lacks sufficient discussion on potential barriers to scaling the intervention beyond the pilot settings. A detailed plan for addressing resource constraints in diverse healthcare systems could enhance its utility.

Response: Many thanks for your helpful comments. Our rigorous methods for intervention design have aimed to maximise the feasibility of the intervention using realist review methodology, co-design approach, PPIE and implementation theory. Intervention development included key stakeholders who are likely to implement and benefit from the intervention in real life who identified a number of potential barriers for routine implementation at the outset which were considered and addressed in the design stage. The intervention brings together resources, including the IMPACT tool and patient information leaflets that are freely available to all health care professionals in the UK. We have also ensured that any prescribing health care professional can conduct the SMR process and the MDT approach enables other HCPs to be involved as and when needed. The intervention aims to make the SMR process more efficient and standardized within the existing systems, saving time and resources. We have made this point clearer in the discussion on pages 20-21.

We have just completed a feasibility study testing the intervention (as we refer to in the discussion) and we have explored the barriers and facilitators for delivering the intervention in routine clinical practice and these will be reported separately in an upcoming paper detailing the feasibility and acceptability of the intervention. This paper will also report the results of our health economic evaluation, and address the possible resource constraints and other potential barriers to scaling the intervention beyond the feasibility study setting.

Reviewer 1: Stakeholder diversity in development process: The study involved stakeholders, including patients, caregivers, and healthcare professionals, but the representation of certain groups, such as minority ethnic populations or those from varied socioeconomic backgrounds, is unclear. Expanding on the demographic diversity of participants could strengthen the intervention's generalizability.

Response: In the methods on page 4 we have added that the patients, carers and HCPs participating in the qualitative study have been recruited from areas with a range of socio-economic deprivation levels, both rural and urban.

In the discussion on page 22 we had already acknowledged that the lack of ethnic and cultural diversity of our sample is a limitation, but we have now added that our PPI group was diverse in terms of gender, age and ethnicity, and their input was included at all stages of the intervention development process, and in the current feasibility study.

Reviewer 1: Outcome measures and feasibility of follow-up plans: Although the manuscript emphasizes tailored follow-up plans for continuity of care, it does not specify measurable outcomes for evaluating the effectiveness of these follow-ups. Including specific metrics, such as adherence rates or health outcomes post-intervention, would provide a more robust evaluation framework.

Response: Many thanks for your comments. As outlined above, the outcome measures for evaluating the effectiveness of the intervention will be reported and discussed in a separate publication based on the feasibility study that we have just completed. The outcome data is currently being analysed.

We have added a section on our outcome measures in the discussion on page 22.

The outcome measures were discussed and agreed with our PPI group and academic and clinical stakeholders. The outcome measures chosen for evaluation in the feasibility study included: medication-related outcomes (number of medications pre and post intervention, types of deprescribing recommendations), clinical-related outcomes (including treatment burden, frailty status, falls), quality of life, healthcare use and cost related outcomes, and safety outcomes (hospitalization and mortality rates).

Reviewer 1: Ethical considerations in data sharing: The manuscript mentions that anonymized qualitative data is available upon request but does not elaborate on the ethical safeguards for participant confidentiality. Further clarity on how sensitive patient data will be protected during broader implementation is necessary.

Response: Thank you for your helpful comments here. The qualitative data is all anonymized to protect patient confidentiality, with the use of participant ID numbers and the removal of any identifiable data to safeguard participant confidentiality. Participants have consented to their data being used to support research in the future, and to their data being shared anonymously with other researchers. Requests for data would be reviewed on a case-by-case basis, to ensure that the purpose of the request is relevant and that the requester has appropriate processes for data management in place.

When the intervention is implemented in routine clinical practice the normal health service rules and regulations would apply to patient data, as they would currently when a health care professional is conducting an SMR, as this would not involve research data.

Reviewer 1: Behaviour change model justification: The manuscript employs the Behaviour Change Wheel (BCW) and Normalization Process Theory (NPT) frameworks but does not sufficiently justify why these models were chosen over others. A comparative discussion could enhance understanding of their appropriateness for this intervention.

Response: We have now included a more explicit justification of our theoretical approach to enhance understanding of their appropriateness for this intervention. Please see the discussion on pages 21-22.

Reviewer 1: Evaluation of deprescribing tools: While the use of tools like the IMPACT tool is highlighted, there is limited discussion on their limitations or feedback from healthcare professionals during the stakeholder workshops. Addressing these could inform future iterations of the intervention.

Response: As part of our feasibility study we conducted a qualitative process evaluation, including health care professional (HCP) interviews providing feedback on the IMPACT tool. This data is currently being analysed and will be written up as a separate paper, as detailed above. The focus of the current manuscript is on intervention development. Some feedback on the IMPACT tool from HCPs in the stakeholder workshops has been reported in table 4, however we have now added more detail on this feedback in table 4, on page 17.

Reviewer 1: Sustainability and long-term implementation: The manuscript outlines a structured intervention development process but lacks a clear strategy for long-term sustainability. Details on funding models, policy alignment, or integration into existing healthcare workflows would strengthen the argument for its widespread adoption.

Response: Thank you for your comments.

We have added more on the long term-sustainability, policy alignment and integration into existing healthcare workflows in the discussion on page 21. For example our intervention is very much in alignment with UK policy, advocating the role of clinical pharmacists within a multidisciplinary approach to SMR in primary care to address inappropriate polypharmacy and in alignment with wider UK policy aims to address overprescribing. We have also added that we are currently conducting policy engagement work to consult with policymakers to identify facilitators to implementation and integration of the intervention within the workflow of primary care teams, but also beyond to community settings.

The focus of this paper is on the development of the intervention, however our feasibility study results which includes a health economic evaluation (as referred to in the discussion) will be published separately. This will discuss further the issues of long term-sustainability, policy alignment and integration into existing healthcare workflows.

Reviewer #2

This is a very interesting paper with some highly useful recommendations for improving the process of SMRs particularly in those with multimorbidity and polypharmacy. It is well written, the qualitative evidence gathered to support the findings is robust and recommendations are sound. I enjoyed reading the paper.

Response: Thank you for your positive feedback.

Reviewer 2: The issues around barriers and facilitators in SMRs have recently been reviewed extensively in another PLOS One paper - see Abuzour AS, Wilson SA, Woodall AA, Mair FS, Clegg A, Shantsila E, Gabbay M, Abaho M, Aslam A, Bollegala D, Cant H, Griffiths A, Hama L, Leeming G, Lo E, Maskell S, O'Connell M, Popoola O, Relton S, Ruddle RA, Schofield P, Sperrin M, Staa TV, Buchan I, Walker LE. A qualitative exploration of barriers to efficient and effective structured medication reviews in primary care: Findings from the DynAIRx study. PLoS One. 2024 Aug 30;19(8):e0299770. doi: 10.1371/journal.pone.0299770. PMID: 39213435; PMCID: PMC11364411, some comparison and reflection between the findings of this team and those of the Southampton group would be helpful.

Response: Thank you for bringing this very relevant paper to our attention. Many of the findings from our qualitative study and realist review are in line with the findings reported in this paper.

We have now referred to the paper in the introduction on page 3 and in the discussion on page 22 and 23.

Reviewer 2: Also, opportunities to use decision support tools (eg AI) in supporting SMRs are highly relevant and while not explored directly in this paper, some discussion of such methodology in the discussion section should be made I feel to reflect this growing area of research interest and practice in the discussion section.

Response: Thank you, this is a very useful suggestion. We have added this to the discussion on page 20/21.

Overall however I feel this is a highly valuable addition to the research evidence in this field. Thank you for your positive comments.

Thank you both for your time and consideration of this submission. We look forward to the opportunity to contribute to the growing body of knowledge on polypharmacy and optimizing medication management in frail older adults, a pressing healthcare challenge with important implications for patient safety and quality of life.

Yours faithfully

Dr Eloise Radcliffe and Dr Kinda Ibrahim

On behalf of all the authors

---

## [Editor Report · Decision Letter 1]

5 Feb 2025

Development of a complex multidisciplinary medication review and deprescribing intervention in primary care for older people living with frailty and polypharmacy

PONE-D-24-53398R1

Dear Dr. Radcliffe,

We’re pleased to inform you that your manuscript has been judged scientifically suitable for publication and will be formally accepted for publication once it meets all outstanding technical requirements.

Kind regards,

Kathleen Bennett

Academic Editor

PLOS ONE
---

## [Editor Report · Acceptance letter]

PONE-D-24-53398R1

PLOS ONE

Dear Dr. Radcliffe,

I'm pleased to inform you that your manuscript has been deemed suitable for publication in PLOS ONE. Congratulations! Your manuscript is now being handed over to our production team.

Kind regards,

on behalf of

Professor Kathleen Bennett

%CORR_ED_EDITOR_ROLE%

PLOS ONE